# VARIATIONAL AUTOENCODERS FOR OPPONENT MODELING IN MULTI-AGENT SYSTEMS

## ABSTRACT

Multi-agent systems exhibit complex behaviors that emanate from the interactions of multiple agents in a shared environment. In this work, we are interested in controlling one agent in a multi-agent system and successfully learn to interact with the other agents that have fixed policies. Modeling the behavior of other agents (opponents) is essential in understanding the interactions of the agents in the system. By taking advantage of recent advances in unsupervised learning, we propose modeling opponents using variational autoencoders. Additionally, many existing methods in the literature assume that the opponent models have access to opponent's observations and actions during both training and execution. To eliminate this assumption, we propose a modification that attempts to identify the underlying opponent model, using only local information of our agent, such as its observations, actions, and rewards. The experiments indicate that our opponent modeling methods achieve equal or greater episodic returns in reinforcement learning tasks against another modeling method.

## 1 INTRODUCTION

In recent years, several promising works (Mnih et al., 2015; Schulman et al., 2015a; Mnih et al., 2016) have arisen in deep reinforcement learning (RL), leading to fruitful results in single-agent scenarios. In this work, we are interested in using single-agent RL in multi-agent systems, where we control one agent and the other agents (opponents) in the environment have fixed policies. The agent should be able to successfully interact with a diverse set of opponents as well as generalize to new unseen opponents. One effective way to address this problem is opponent modeling. The opponent models output specific characteristics of the opponents based on their trajectories. By successfully modeling the opponents, the agent can reason about opponents' behaviors and goals and adjust its policy to achieve the optimal outcome. There is a rich literature of modeling opponents in the multi-agent systems (Albrecht & Stone, 2018).

Several recent works have proposed learning opponent models using deep learning architectures (He et al., 2016; Raileanu et al., 2018; Grover et al., 2018a; Rabinowitz et al., 2018). In this work, we focus on learning opponent models using Variational Autoencoders (VAEs) (Kingma & Welling, 2014). This work is, to the best of our knowledge, the first attempt to use VAEs in multi-agent scenarios. VAE are generative models that are commonly used for learning representations of the data, and various works use them in RL for learning representations of the environment (Igl et al., 2018; Ha & Schmidhuber, 2018; Zintgraf et al., 2019). We first propose a VAE for learning opponent representations in multi-agent systems based on the opponent trajectories.

A shortcoming of this approach and most opponent modeling methods, as will be presented in Section 2, is that they require access to opponent's information, such as observations and actions, during training as well as execution. This assumption is too limiting in the majority of scenarios. For example, consider Poker, where each agent never has access to the opponent's observations. Nevertheless, during Poker, humans can reason about the opponent's behaviors and goals using only their local observations. For example, an increase in the table's pot could mean that the opponent either holds strong cards or is bluffing. Based on the idea that an agent can reason about an opponent's model using its observations, actions, and rewards in a recurrent fashion, we propose a second VAE-based architecture. The encoder of the VAE learns to represent opponents' models conditioned on only local information removing the requirement to access the opponents' information during execution.

To summarize our contribution, in this work, we explore VAEs for opponent modeling in multi-agent systems. We are not interested in VAEs as generative models but as methods for learning representations. We evaluate our proposed methodology using a toy example and the commonly used Multi-agent Particle Environment (Mordatch & Abbeel, 2017). We evaluate the quality of the learned representations, and the episodic returns that RL algorithms can achieve. The experiments indicate that opponent modeling without opponents' information can perform the same or even better in RL compared to models that access the opponent's information.

## 2 RELATED WORK

**Learning Opponent Models.** In this work, we are interested in opponent modeling methods that use neural networks to learn representations of the opponents. He et al. (2016) proposed an opponent modeling method that learns a modeling network to reconstruct the opponent's actions given the opponent observations. Raileanu et al. (2018) developed an algorithm for learning to infer opponents' goals using the policy of the controlled agent. Grover et al. (2018a) proposed an encoder-decoder method for modeling the opponent's policy. The encoder learns a point-based representation of different opponents' trajectories, and the decoder learns to reconstruct the' opponent's policy given samples from the embedding space. Additionally, Grover et al. (2018a) introduce an objective to separate embeddings of different agents into different clusters.

$$d(z_+, z_-, z) = \frac{1}{(1 + e^{|z-z_-|_2 - |z-z_+|_2})^2} \tag{1}$$

where $z_+$ and $z$ are embeddings of the same agent from two different episodes and embedding $z_-$ is generated from the episode of a different agent. Rabinowitz et al. (2018) proposed the Theory of mind Network (TomNet), which learns embedding-based representations of opponents for meta-learning. Tacchetti et al. (2018) proposed RFM to model opponents using graph neural networks. A common assumption among these methods, that this work aims to eliminate, is that access to opponents trajectories is available during execution.

**Representation Learning in Reinforcement Learning.** Another topic that has received significant attention, recently, is representation learning in RL. Using unsupervised learning techniques to learn low dimensional representations of the MDP has led to significant improvement in RL. Ha & Schmidhuber (2018) proposed a VAE-based and a forward model to learn state representations of the environment. Hausman et al. (2018) learned tasks embeddings and interpolated them to solve harder tasks. Igl et al. (2018) used a VAE for learning representation in partially-observable environments. Gupta et al. (2018) proposed MAESN, which learns Gaussian embeddings to represent different tasks during meta-training and manages to quickly adapt to new task during meta-testing. The work of Zintgraf et al. (2019) is closely related, where Zintgraf et al. proposed a recurrent VAE model, which receives as input the observation, action, reward of the agent, and learns a variational distribution of tasks. Rakelly et al. (2019) used representations from an encoder for off-policy meta-RL. Note that, all these works have been applied for learning representations of tasks or properties of the environments. On the contrary, our approach is focused on learning representations of the opponents.

## 3 BACKGROUND

### 3.1 REINFORCEMENT LEARNING

Markov Decision Processes (MDPs) are commonly used to model decision making problems. An MDP consists of the set of states $\mathbb{S}$, the set of actions $\mathbb{A}$, the transition function, $P(s'|s, a)$, which is the probability of the next state, $s'$, given the current state, $s$, and the action, $a$, and the reward function, $r(s', a, s)$, that returns a scalar value conditioned on two consecutive states and the intermediate action. A policy function is used to choose an action given a state, which can be stochastic $a \sim \pi(a|s)$ or deterministic $a = \mu(s)$. Given a policy $\pi$, the state value function is defined as $V(s_t) = \mathbb{E}_\pi[\sum_{i=t}^{H} \gamma^{i-t} r_t | s = s_t]$ and the state-action value (Q-value) $Q(s_t, a_t) = \mathbb{E}_\pi[\sum_{i=t}^{H} \gamma^{i-t} r_t | s = s_t, a = a_t]$, where $0 \leq \gamma \leq 1$ is the discount factor and $H$ is the finite horizon of the episode. The goal of RL is to compute the policy that maximizes state value function $V$, when the transition and the reward functions are unknown.

There is a large number of RL algorithms; however, in this work, we focus on two actor-critic algorithms; the synchronous Advantage Actor-Critic (A2C) (Mnih et al., 2016; Dhariwal et al., 2017) and the Deep Deterministic Policy Gradient (DDPG) (Silver et al., 2014; Lillicrap et al., 2015). DDPG is an off-policy algorithm, using an experience replay for breaking the correlation between consecutive samples and target networks for stabilizing the training (Mnih et al., 2015). Given an actor network with parameters $\boldsymbol{\theta}$ and a critic network with parameter $\phi$, the gradient updates are performed using the following equations.

$$\min_{\boldsymbol{\phi}} \frac{1}{2}\mathbb{E}_B[(r + \gamma \cdot Q_{\text{target},\boldsymbol{\phi}'}(\boldsymbol{s}', \mu_{\text{target},\boldsymbol{\theta}'}(\boldsymbol{s}')) - Q_{\boldsymbol{\phi}}(\boldsymbol{s}, \boldsymbol{a}))^2]$$
$$\min_{\boldsymbol{\theta}} -\mathbb{E}_B[Q_{\phi}(\boldsymbol{s}, \mu_{\boldsymbol{\theta}}(\boldsymbol{s}))] \tag{2}$$

On the other hand, A2C is an on-policy actor-critic algorithm, using parallel environments to break the correlation between consecutive samples. The actor-critic parameters are optimized by:

$$\min_{\boldsymbol{\theta},\boldsymbol{\phi}} \mathbb{E}_B[-\hat{A}\log \pi_\theta(\boldsymbol{a}|\boldsymbol{s}) + \frac{1}{2}(r + \gamma V_\phi(\boldsymbol{s}') - V_\phi(\boldsymbol{s}))^2] \tag{3}$$

where the advantage term, $\hat{A}$, can be computed using the Generalized Advantage Estimation (GAE) (Schulman et al., 2015b).

## 3.2 Variational Autoencoders

Consider samples from a dataset $\boldsymbol{x} \in \mathbb{X}$ that are generated from some hidden (latent) random variable $\boldsymbol{z}$ based on a generative distribution $p_{\boldsymbol{u}}(\boldsymbol{x}|\boldsymbol{z})$ with unknown parameter $\boldsymbol{u}$ and a prior distribution on the latent variables, which we assume is a Gaussian with $\boldsymbol{0}$ mean and unit variance $p(\boldsymbol{z}) = \mathcal{N}(\boldsymbol{z}; \boldsymbol{0}, \boldsymbol{I})$. We are interested in approximating the true posterior $p(\boldsymbol{z}|\boldsymbol{x})$ with a variational parametric distribution $q_{\boldsymbol{w}}(\boldsymbol{z}|\boldsymbol{x}) = \mathcal{N}(\boldsymbol{z}; \boldsymbol{\mu}, \boldsymbol{\Sigma}, \boldsymbol{w})$. Kingma & Welling (2014) proposed the Variational Autoencoders (VAE) to learn this distribution. Starting from the Kullback-Leibler (KL) divergence from the approximate to the true posterior $D_{\text{KL}}(q_{\boldsymbol{w}}(\boldsymbol{z}|\boldsymbol{x})\|p(\boldsymbol{z}|\boldsymbol{x}))$, the lower bound on the evidence $\log p(\boldsymbol{x})$ is derived as:

$$\log p(\boldsymbol{x}) \geq \mathbb{E}_{\boldsymbol{z} \sim q_{\boldsymbol{w}}(\boldsymbol{z}|\boldsymbol{x})}[\log p_{\boldsymbol{u}}(\boldsymbol{x}|\boldsymbol{z})] - D_{\text{KL}}(q_{\boldsymbol{w}}(\boldsymbol{z}|\boldsymbol{x})\|p(\boldsymbol{z})) \tag{4}$$

The architecture consists of an encoder which receives a sample $\boldsymbol{x}$ and generates the Gaussian variational distribution $p(\boldsymbol{z}|\boldsymbol{x}; \boldsymbol{w})$. The decoder receives a sample from the Gaussian variational distribution and reconstructs the initial input $\boldsymbol{x}$. The architecture is trained using the reparameterization trick Kingma & Welling (2014). Higgins et al. (2017) proposed $\beta$-VAE, where a parameter $\beta \geq 0$ is used to control the trade-off between the reconstruction loss and the KL-divergence.

$$L(\boldsymbol{x}; \boldsymbol{w}, \boldsymbol{v}) = \mathbb{E}_{\boldsymbol{z} \sim q_{\boldsymbol{w}}(\boldsymbol{z}|\boldsymbol{x})}[\log p_{\boldsymbol{u}}(\boldsymbol{x}|\boldsymbol{z})] - \beta D_{\text{KL}}(q_{\boldsymbol{w}}(\boldsymbol{z}|\boldsymbol{x})\|p(\boldsymbol{z})) \tag{5}$$

# 4 Approach

## 4.1 Problem Formulation

We consider a modified Markov Game (Littman, 1994), which consists of $N$ agents $\mathbb{I} = \{1, 2, ..., N\}$, the set of states $\mathbb{S}$, the set of actions $\mathbb{A} = \mathbb{A}_1 \times \mathbb{A}_{-1}$, the transition function $P : \mathbb{S} \times \mathbb{A} \times \mathbb{S} \to \mathbb{R}$ and the reward function $r : \mathbb{S} \times \mathbb{A} \times \mathbb{S} \to \mathbb{R}^N$. We consider partially observable settings, where each agent $i$ has access only to its local observation $\boldsymbol{o}_i$ and reward $r_i$. Additionally, two sets of pretrained opponents are provided $\mathbb{T} = \{\mathbb{I}_{-1,m}\}_{m=1}^{m=M}$ and $\mathbb{G} = \{\mathbb{I}_{-1,m}\}_{m=1}^{m=M}$, which are responsible for providing the joint action $\mathbb{A}_{-1}$. Note that by opponent we refer to $\mathbb{I}_{-1,m}$, which consists of one or more agents, independently from the type of the interactions (cooperative, mixed or competitive). At the beginning of each episode, we sample a pretrained opponent from the set $\mathbb{T}$ during training or from $\mathbb{G}$ during testing. Our goal is to train the agent 1 using RL, to maximize the average return against opponents from the training set, $\mathbb{T}$, and generalize to opponents sampled from the test $\mathbb{G}$. Note, that when we refer to agent 1 we drop the subscript.

$$\max \mathbb{E}_\pi[\mathbb{E}_\mathbb{T}[\sum_t \gamma^t r_t]] \tag{6}$$

## 4.2 VARIATIONAL AUTOENCODER WITH ACCESS TO OPPONENT'S INFORMATION

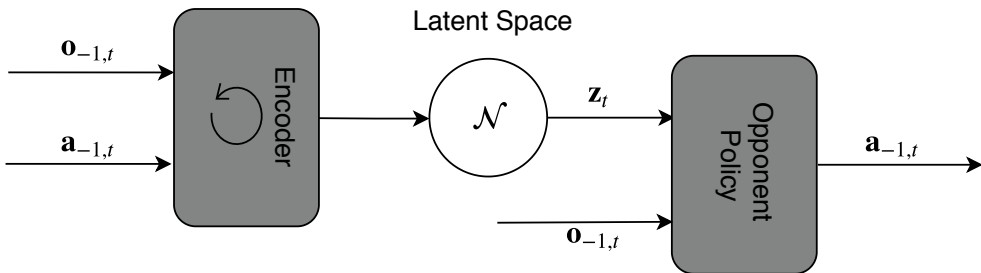

Figure 1: Diagram of the proposed VAE architecture

We assume a number of $K$ provided episode trajectories for each pretrained opponents $j \in \mathbb{T}$, $E^{(j)} = \{\boldsymbol{\tau}_{-1}^{(j,k)}\}_{k=0}^{k=K-1}$, where $\boldsymbol{\tau}_{-1}^{(j,k)} = \{\boldsymbol{o}_{-1,t}, \boldsymbol{a}_{-1,t}\}_{t=0}^{t=H}$, and $\boldsymbol{o}_{-1,t}, \boldsymbol{a}_{-1,t}$ are the observations and actions of the opponent at the time step $t$ in the trajectory. These trajectories are generated from the opponents in set $\mathbb{T}$, which are represented in the latent space from the variable $\boldsymbol{z}$ and for which we assume there exists an unknown model $p_{\boldsymbol{u}}(\boldsymbol{\tau}_{-1}|\boldsymbol{z})$. Our goal is to approximate the unknown posterior, $p(\boldsymbol{z}|\boldsymbol{\tau}_{-1})$, using a variational Gaussian distribution $\mathcal{N}(\boldsymbol{\mu}, \boldsymbol{\Sigma}; \boldsymbol{w})$ with parameters $\boldsymbol{w}$. We consider using a $\beta$-VAE for the sequential task:

$$L(\boldsymbol{\tau}_{-1}; \boldsymbol{w}, \boldsymbol{u}) = \mathbb{E}_{z \sim q_{\boldsymbol{w}}(\boldsymbol{z}|\boldsymbol{\tau}_{-1})}[\log p_{\boldsymbol{u}}(\boldsymbol{\tau}_{-1}|\boldsymbol{z})] - \beta D_{\mathrm{KL}}(q_{\boldsymbol{w}}(\boldsymbol{z}|\boldsymbol{\tau}_{-1})\|p(\boldsymbol{z})) \qquad (7)$$

We can further subtract the discrimination objective (equation 1) that was proposed by Grover et al. (2018a). Since the discrimination objective is always non-negative, we derive and optimize a lower bound as:

$$\begin{aligned} L(\boldsymbol{\tau}_{-1}; \boldsymbol{w}, \boldsymbol{u}) \geq \; &\mathbb{E}_{z \sim q_{\boldsymbol{w}}(\boldsymbol{z}|\boldsymbol{\tau}_{-1})}[\log p_{\boldsymbol{u}}(\boldsymbol{\tau}_{-1}|\boldsymbol{z})] \\ &- \beta D_{\mathrm{KL}}(q_{\boldsymbol{w}}(\boldsymbol{z}|\boldsymbol{\tau}_{-1})\|p(\boldsymbol{z})) - \lambda d(\mathbb{E}(\boldsymbol{z}_+), \mathbb{E}(\boldsymbol{z}_-), \mathbb{E}(\boldsymbol{z})) \end{aligned} \qquad (8)$$

The discrimination objective receives as input the mean of the variational Gaussian distribution, produced by three different trajectories. Despite the less tight lower bound, the discrimination objective will separate the opponents in the embedding space, which could potentially lead to higher episodic returns. At each time step $t$, the recurrent encoder network generates a latent sample $z_t$, which is conditioned on the opponent's trajectory $\boldsymbol{\tau}_{-1,:t}$, until this time step. The KL divergence can be written as:

$$D_{\mathrm{KL}}(q_{\boldsymbol{w}}(\boldsymbol{z}|\boldsymbol{\tau}_{-1})\|p(\boldsymbol{z})) = \sum_{t=1}^{H} D_{\mathrm{KL}}(q_{\boldsymbol{w}}(\boldsymbol{z}_t|\boldsymbol{\tau}_{-1,:t})\|p(\boldsymbol{z}_t)) \qquad (9)$$

The lower bound consist of the reconstruction loss of the trajectory which involves the observation and actions of the opponent. The opponent's action, at each time step depends on its observation and the opponent's policy, which is represented by the latent variable $z$. We use a decoder that consists of fully-connected layers, however, a recurrent network can be used, if we instead assume that the opponent decides its actions based on the history of its observations. Additionally, the observation at each time step depends only on the dynamics of the environment and the actions of the agents and not on the identity of the opponent. Therefore, the reconstruction loss factorizes as:

$$\begin{aligned} \log p_{\boldsymbol{u}}(\boldsymbol{\tau}_{-1}|\boldsymbol{z}) &= \sum_{t=1}^{H} \log p_{\boldsymbol{u}}(\boldsymbol{a}_{-1,t}|\boldsymbol{o}_{-1,t}, \boldsymbol{z}_t) p_{\boldsymbol{u}}(\boldsymbol{o}_{-1,t}|\boldsymbol{o}_{t-1}, \boldsymbol{o}_{-1,t-1}, \boldsymbol{a}_{t-1}, \boldsymbol{a}_{-1,t-1}) \\ &\propto \sum_{t=1}^{H} \log p_{\boldsymbol{u}}(\boldsymbol{a}_{-1,t}|\boldsymbol{o}_{-1,t}, \boldsymbol{z}_t) \end{aligned} \qquad (10)$$

From the equation above, we observe that the loss is the reconstruction of the opponent's policy given the current observation and a sample from the latent variable. Overall, our proposed VAE takes the form of a Conditional VAE (Sohn et al., 2015). Figure 1 illustrates the diagram of the VAE. The full pseudocode of the method is provided in the Appendix D.

### 4.3 VARIATIONAL AUTOENCODER WITHOUT ACCESS TO OPPONENT'S INFORMATION

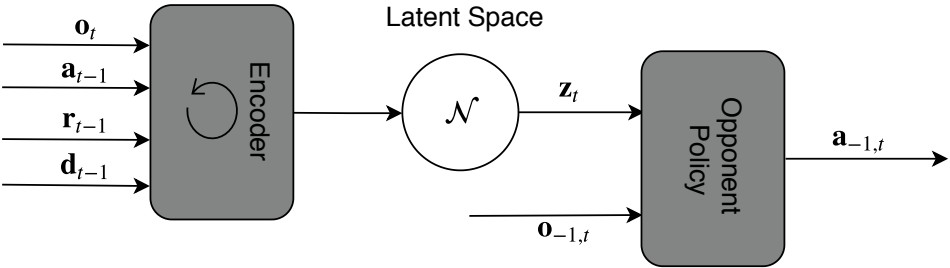

Figure 2: Diagram of the proposed VAE architecture using local information

In Sections 1 and 2, it was noted that most agent modeling methods assume access to opponent's observations and actions is available both during training and execution. To eliminate this assumption, we propose a VAE that uses a parametric variational distribution which is conditioned on the observation-action-reward triplet of the agent that we control and a variable $d$ indicating whether the episode has terminated; $q_{\boldsymbol{w}}(\boldsymbol{z}|\boldsymbol{\tau} = (\boldsymbol{o}, \boldsymbol{a}, r, d))$. More precisely, our goal is to approximate the true posterior that is conditioned on opponent's information, with a variational distribution that only depends on local information. The use of this local information in a recurrent fashion has been successfully used in meta-RL settings (Wang et al., 2016; Duan et al., 2016). We start by computing the KL divergence between the two distributions:

$$D_{\mathrm{KL}}(q_{\boldsymbol{w}}(\boldsymbol{z}|\boldsymbol{\tau})\|p(\boldsymbol{z}|\boldsymbol{\tau}_{-1})) = \mathbb{E}_{\boldsymbol{z}\sim q_{\boldsymbol{w}}(\boldsymbol{z}|\boldsymbol{\tau})}[\log q_{\boldsymbol{w}}(\boldsymbol{z}|\boldsymbol{\tau}) - \log p(\boldsymbol{z}|\boldsymbol{\tau}_{-1})] \tag{11}$$

By following the works of Kingma & Welling (2014) and Higgins et al. (2017) and using the Jensen inequality, the VAE objective can be written as:

$$L(\boldsymbol{\tau}, \boldsymbol{\tau}_{-1}; \boldsymbol{w}, \boldsymbol{v}) = \mathbb{E}_{\boldsymbol{z}\sim q_{\boldsymbol{w}}(\boldsymbol{z}|\boldsymbol{\tau})}[\log p_{\boldsymbol{u}}(\boldsymbol{\tau}_{-1}|\boldsymbol{z})] - \beta D_{\mathrm{KL}}(q_{\boldsymbol{w}}(\boldsymbol{z}|\boldsymbol{\tau})\|p(\boldsymbol{z})) \tag{12}$$

The reconstruction loss factorizes exactly similar to equation 10. From equation 12, it can be seen that the variational distribution only depends on locally available information. Since during execution, only the encoder is required to generate the opponent's model, this approach removes the assumption that access to the opponent's observations and actions is available during execution. Figure 2 presents the diagram of the VAE.

### 4.4 REINFORCEMENT LEARNING TRAINING

We use the latent variable $\boldsymbol{z}$ augmented with the agent's observation to condition the policy of our agent, which is optimized using RL. Consider the augmented observation space $\mathbb{O}' = \mathbb{O} \times \mathbb{Z}$, where $\mathbb{O}$ is the original observation space of the our agent in the Markov game, and $\mathbb{Z}$ is the representation space of the opponent models. The advantage of learning the policy on $\mathbb{O}'$ compared to $\mathbb{O}$ is that the policy can adapt to different $\boldsymbol{z} \in \mathbb{Z}$.

After training the variational autoencoder that was described in Section 4.2, we use it to train our agent against the opponents in the set $\mathbb{T}$. We use the DDPG (Lillicrap et al., 2015) algorithm for this task. We did not manage to optimize the representation jointly with the policy, neither with DDPG or A2C. At the beginning of each episode, we sample an opponent from the set $\mathbb{T}$. The agent's input is the local observation and a sample from the variational distribution. We refer to this as OMDDPG (Opponent Modeling DDPG), and the full pseudocode is provided in Appendix D.

We optimize the second proposed VAE method jointly with the policy of the controlled agent. We use the A2C algorithm, similarly to the meta-learning algorithm $RL^2$ (Wang et al., 2016; Duan et al., 2016). In the rest of this paper, we refer to this as SMA2C (Self Modeling A2C). The actor's and the critic's input is the local observation and a sample from the latent space. We back-propagate the gradient from both the actor and the critic loss to the parameters of the encoder. Therefore, the encoder's parameters are shaped to maximize both the VAE's objective as well as the discounted sum of rewards. The full pseudocode is provided in Appendix D.

## 5 EXPERIMENTS

### 5.1 TOY EXAMPLE

We will first provide a toy example to illustrate SMA2C. We consider the classic repeated game of prisoner's dilemma with a constant episode length of 25 time steps. We control one agent, and the other agent is selected randomly between two possible opponent policies. The first opponent always defects, while the second opponent follows a tit-for-tat policy. At the beginning of the episode, one of the two opponents is randomly selected. We train SMA2C against the two possible opponents. The agent that we control has to identify the correct opponent, and the optimal policy, it can achieve, is to defect against opponent one and collaborate with opponent two. Figure 3 shows the payoff matrix, the embedding space at the last time step of the episode, and the episodic return that SMA2C and A2C achieve during training. Note that, based on the payoff matrix, the optimal average episodic return that can be achieved is $24.5$.

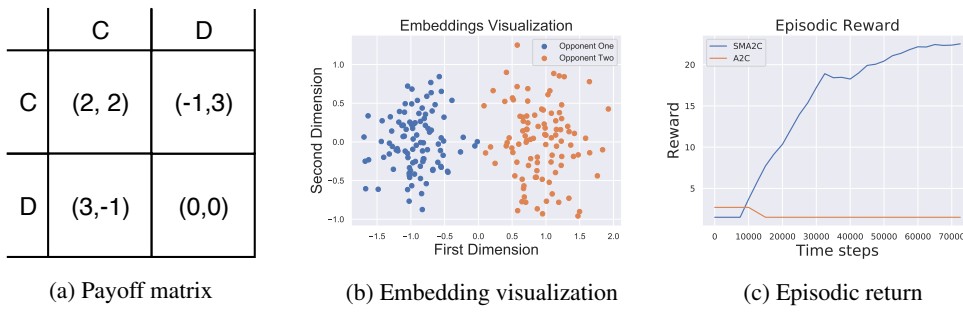

|   | C | D |
|---|---|---|
| C | (2, 2) | (-1,3) |
| D | (3,-1) | (0,0) |

(a) Payoff matrix      (b) Embedding visualization      (c) Episodic return

Figure 3: Payoff matrix, embedding visualization and episodic return during training.

### 5.2 EXPERIMENTAL FRAMEWORK

To evaluate the proposed methods in more complex environments, we used the Multi-agent Particle Environment (MPE) (Mordatch & Abbeel, 2017), which provides several different multi-agent environments. The environments have continuous observation, discrete action space, and fixed-length episodes of 25 time steps. Four environments are used for evaluating the proposed methodology; the speaker-listener, the double-speaker listener, the predator-prey, and the spread. In Appendix A, descriptions of the different environments are provided. During the experiments, we evaluated the two proposed algorithms OMDDPG and SMA2C as well as the modeling method of Grover et al. (2018a) combined with DDPG (Lillicrap et al., 2015).

In all the environments, we pretrain ten different opponents, where five are used for training and five for testing. In the speaker-listener environment, we control the listener, and we create ten different speakers using different communication messages for different colors. In the double speaker-listener, which consists of two agents that have to be both listener and speaker simultaneously, we control the first agent. We create a diverse set of opponents that have different communication messages similar to speaker-listener, while they learn to navigate using the MADDPG algorithm (Lowe et al., 2017), with different initial random seeds. In the predator-prey environment, we control the prey and pretrain the three other agents in the environment using MADDPG with different initial parameters. Similarly, in spread, we control one of the agents, while the opponents are pretrained using MADDPG.

We use agent generalization graphs (Grover et al., 2018b) to evaluate the generalization of the proposed methods. We evaluate two types of generalizations in this work. First, we evaluate the episodic returns against the opponents that are used for training, $\mathbb{T}$, which Grover et al. (2018b) call "weak generalization". Secondly, we evaluate against unknown opponents from set $\mathbb{G}$, which is called "strong generalization". A figure of an agent generalization graph is provided in Appendix E.

### 5.3 EVALUATION OF THE REPRESENTATIONS

To evaluate the representations created from our models we will estimate the Mutual Information (MI) between the variational distribution ($q(z|\tau)$ or $q(z|\tau_{-1})$) and the prior on the opponents' identities, which is uniform. This is a common method to estimate the quality of the representation (Chen et al., 2018; Hjelm et al., 2019). To estimate the MI, we use the Mutual Information Neural Estimation (MINE) (Belghazi et al., 2018). Note that, the upper bound of the MI, the entropy of the uniform distribution, in our experiments is $1.61$. We gather 200 trajectories against each opponent in $\mathbb{T}$, where $80\%$ of them are used for training and the remaining for testing. The visualization of the embedding space, for the predator-prey environment, is provided in Appendix C.

Table 1: MI estimations using MINE in the double speaker-listener and the predator-prey environment of the embeddings at the 15th, 20th and 25th time step of the trajectory.

| Algorithm | Double speaker-listener | | | Predator-prey | | |
|---|---|---|---|---|---|---|
| | 15 | 20 | 25 | 15 | 20 | 25 |
| OMDDPG | 0.86 | 0.81 | 0.86 | 0.87 | 0.86 | 0.85 |
| SMA2C | 0.73 | 0.72 | 0.7 | 1.27 | 1.04 | 1.39 |
| Grover et al. (2018a) | 1.16 | 1.21 | 1.20 | 1.29 | 1.38 | 1.41 |

From Table 1, we observe that the method of Grover et al. (2018a) achieves significantly higher values of MI. We believe that the main reason behind this is the discrimination objective that implicitly increases MI. This is apparent in the MI values of OMDDPG as well. SMA2C manages to create opponent representations, based only on the local information of our agent, that have information about the opponent identities. Additionally, based on Figure 4, we observe that the value of MI is not directly related to the episodic returns in RL tasks. In Appendix B, we demonstrate that when we detach the encoder's parameters from the policy optimization, the MI decreases.

### 5.4 REINFORCEMENT LEARNING PERFORMANCE

We evaluate the proposed opponent modeling methods in RL settings. In Figure 4, the episodic returns for the three methods in all four environments are presented.

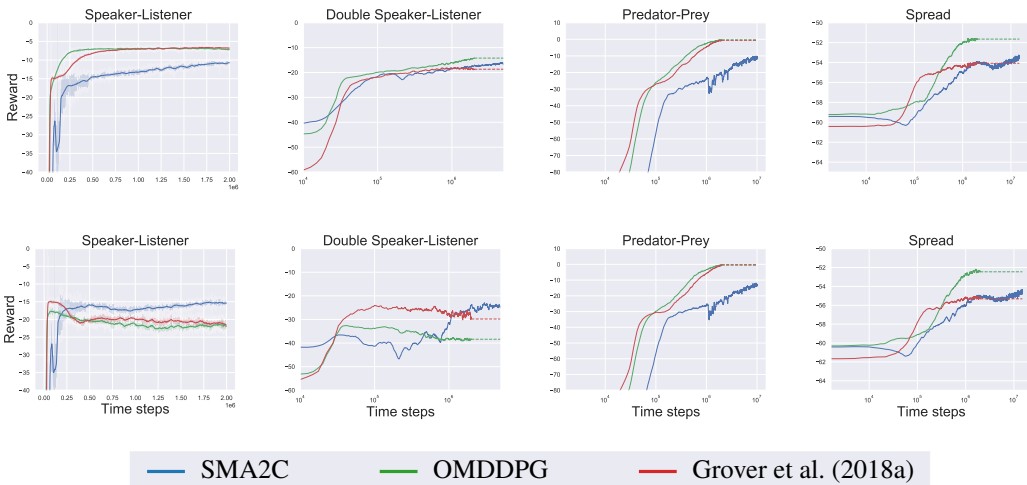

Figure 4: Episodic return during training against the opponents from $\mathbb{T}$ (top row) and $\mathbb{G}$ (bottom row). Four environments are evaluated; speaker-listener (first column), double speaker-listener (second column), predator-prey (third column) and spread (fourth column). For the double speaker-listener, predator-prey and spread environments the x-axis is in logarithmic scale.

Every line corresponds to the average return over five runs with different initial seeds, and the shadowed part represents the $95\%$ confidence interval. We evaluate the models every 1000 episodes for

100 episodes. During the evaluation, we sample an embedding from the variational distribution at each time step, and the agent follows the greedy policy. The hyperparameters for all the experiments in Figure 4 were optimized on weak generalization scenarios, against opponents from set $\mathbb{T}$. Details about the implementation and hyperparameters that were used for generating the figures are presented in Appendix D.

OMDDPG is an upper baseline for SMA2C achieving higher returns in all environments during weak generalization. However, OMDDPG, as well as Grover et al. (2018a), tend to overfit and perform poorly during strong generalization in the speaker-listener and double speaker-listener environment. SMA2C achieves higher returns that Grover et al. (2018a) in more than half of the scenarios. Below, in the Section 5.5, we perform an ablation study on different inputs in the encoder of SMA2C. In Appendix B, we evaluate whether back-propagating the RL loss to the parameters of the encoder, in SMA2C, affects the episodic returns.

## 5.5 ABLATION STUDY ON SMA2C INPUTS

We perform an ablation study to assess the performance requirements of the SMA2C. Our proposed method utilizes the observation, action, reward, and termination sequence to generate the opponent's model. We use different combinations of these elements in the encoder and compare the average episodic returns. In Figure 5, the average episode return is presented for three different cases; SMA2C with all inputs, SMA2C with only observation and action as inputs and SMA2C with only observation as input; for all four environments.

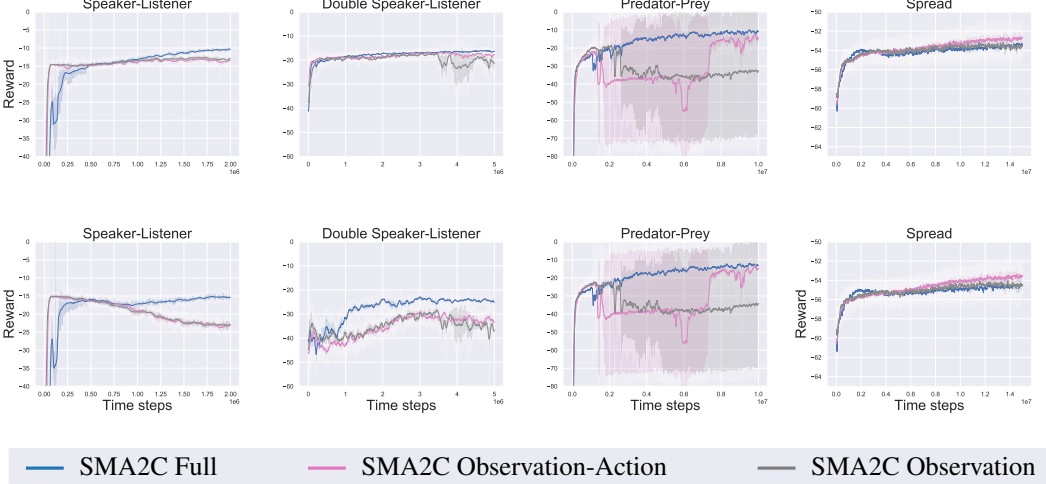

Figure 5: Ablation on the episodic returns for different inputs in the VAE of SMA2C for weak (top row) and strong (bottom row) generalization in all four environments.

## 5.6 ABLATION STUDY ON THE DISCRIMINATION OBJECTIVE

Another element of this work is the utilization of the discrimination objective of the Grover et al. (2018a) in the VAE loss. To better understand how the opponent separation in the embedding space is related to RL performance, below, Figure 6 shows the episodic return during the training for the OMDDPG with and without the discrimination objective is presented. Using the discrimination objective has a significant impact on the episodic returns in the speaker-listener and the double speaker-listener environment.

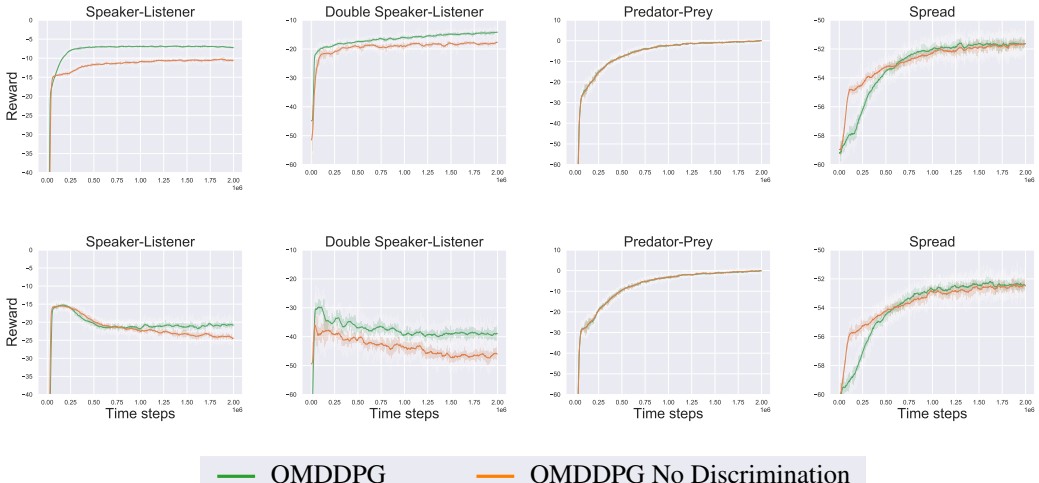

Figure 6: Ablation study on the importance of the discrimination objective. Episodic returns in all four environments for weak (top row) and strong (bottom row) generalization.

## 6 CONCLUSION

To conclude this work, we proposed two methods for opponent modeling in multi-agent systems using variational autoencoders. First, we proposed OMDDPG a VAE-based method that uses the common assumption that access to opponents' information is available during execution. The goal of this work is to motivate opponent modeling without access to opponent's information. The core contribution of this work is SMA2C, which learns representations without requiring access to opponent's information during execution. We performed a thorough experimental evaluation of the proposed methodology. We evaluated the quality of the representations produced by our models as well as the episodic return that can achieve in RL tasks. The experiments conclusively indicate that access to the opponent's information is not necessary during execution, eliminating a long-standing assumption of the prior work. Additionally, we provided evidence that the relationship between the MI and the RL performance is not apparent. In the future, we would like to research how these models can be used for non-stationary opponents. Particularly, there are two scenarios worth investigating; the first is multi-agent deep RL, where different agents are learning concurrently leading to non-stationarity in the environment, which prevents the agents from learning optimal policies. Secondly, we would like to explore whether the proposed models can deal with opponents that try to deceive it and exploit the controlled agent (Ganzfried & Sandholm, 2011; 2015).

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

# A    Details of the Experimental Environment

## A.1    Speaker-Listener Environment

The speaker-listener environment consists of two agents, called speaker and listener as well as three designated landmarks that each one has a different color, red green or blue. At the beginning of the episode, the listener is assigned a color, which can be red, green, or blue. The task of the listener is to navigate to the landmark that has the same color. However, the color of the listener can only be observed by the speaker. So the speaker has to learn to communicate the correct color to the listener. The listener should be able to understand the generated message of the speaker and to navigate to the correct color. The observation space of the listener has 13 dimensions, which consists of the position of the listener, the positions of the three landmarks in the 2D environment and the 5-dimensional communication message of the speaker and its action space has five dimensions. The speaker has a 3-dimensional observation space, which is a vector assigned to the color of the listener, and 5-dimensional action space. We use our method to train the listener to be able to understand a set of different speakers, pretrained speakers that use different communication messages. Both speaker and listener share the same reward, which is the negative Euclidean distance between the listener and the correct landmark. In Figure 7a, an instance from the speaker-listener environment is presented.

## A.2    Double Speaker-Listener Environment

The double speaker-listener environment consists of two agents and three designated landmarks that each one has a different color, red green or blue, similarly to the speaker-listener environment. The only difference is that both agents are simultaneously both speakers and listeners. Therefore, at the beginning of the episode, each agent has a color that can only be observed by the other agent. Each agent must learn both to communicate a message to the other agent as well as navigate to the correct landmark. The observation space of each agent has 16 dimensions, where 13 of them are the same as the listener's in the previous environment, and the other three are the vector assigned to the color of the opponent, while the action space is 5-dimensional for the navigation actions and 5-dimensional for the communication message as well. The reward is the average of the negative Euclidean distances between each agent and the correct landmark. This environment is significantly more difficult compared to the speaker-listener because our agent has to infer both the color that the other agent observes as well as communicate the correct message that the opponent expects. In Figure 7b, an instance from the double speaker-listener environment is presented.

## A.3    Predator-Prey environment

This environment consists of one prey agent and three predator agents. At the beginning of the episode, the prey and the predators are randomly placed on a 2D map. The goal of the prey is to avoid being caught by predators. In the environment, there are additionally two large black obstacles in order to block the agents. The advantage of the prey compared to the three predators is that it can move faster compared to the three adversaries. This environment, compared to the previous two, is competitive. We deliberately chose this environment in order to prove that our method is agnostic to the environment setting. In our work, we will apply the proposed algorithm in the prey agent in order to examine whether it can avoid a large number of different pretrained predator agents. The observation of each agent has 14 dimensions representing the agents' positions as well as the obstacles in the 2D space, while their action space consists of 5 actions. In Figure 7c, an instance from the predator-prey environment is presented.

## A.4    Spread Environment

The spread environment consists of three large agents and three landmarks as well. At the beginning of the episode, the three agents and the three landmarks are spread randomly in the 2D space. The goal of the agents is to navigate to three different landmarks without colliding. The reward is the negative distance of each agent from the landmark. In the case of collision, there is an additional negative reward. The reward is the same for all agents. All the agents have the same observation space, which consists of 18 dimensions, while their action space consists of 5 actions. In Figure 7d, an instance from the spread environment, is presented.

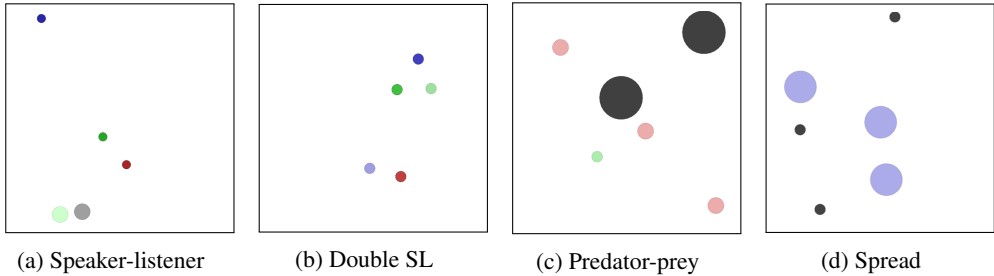

|(a) Speaker-listener|(b) Double SL|(c) Predator-prey|(d) Spread|

Figure 7: Instances of different MPE environments

# B    ABLATION STUDY ON RL BACK-PROPAGATION

We evaluate SMA2C without back-propagating the gradients of the RL loss to the parameters of the encoder. Therefore, the encoder is only trained based on ELBO.

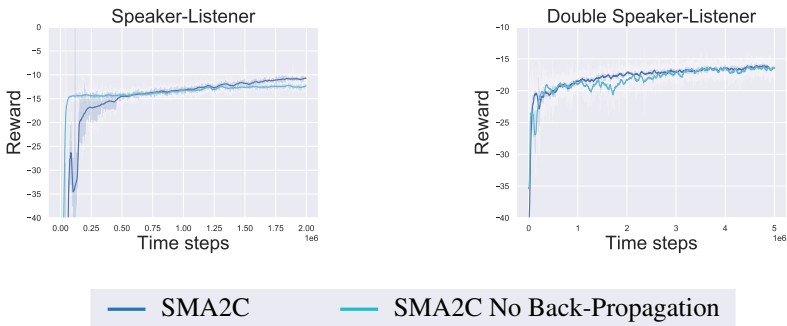

Figure 8: Ablation study on back-propagation of the RL loss to the encoder's parameters for weak generalization.

Figure 8 verifies that not performing back-propagation does not significantly affect the episodic returns that SMA2C achieves during weak generalization. Additionally, we compute the MI between the embeddings and the opponents' identities similarly to Section 5.3 for the double speaker-listener environment. We observe that the MI decreases when we do not perform back-propagation to the embeddings of the encoder.

Table 2: MI estimations using MINE in the double speaker-listener environment of the embeddings at the 15th, 20th and 25th time step of the trajectory.

| | Double speaker-listener | | |
|---|---|---|---|
| **Algorithm** | 15 | 20 | 25 |
| SMA2C | 0.73 | 0.72 | 0.7 |
| SMA2C No Back-propagation | 0.61 | 0.62 | 0.58 |

# C    EMBEDDING VISUALIZATION

Figure 9 visualizes the embedding space for the three different evaluated algorithms in the predator-prey environment. Note that the embeddings were generated using interactions between the opponents from the set $\mathbb{T}$ and the trained agent that we control.

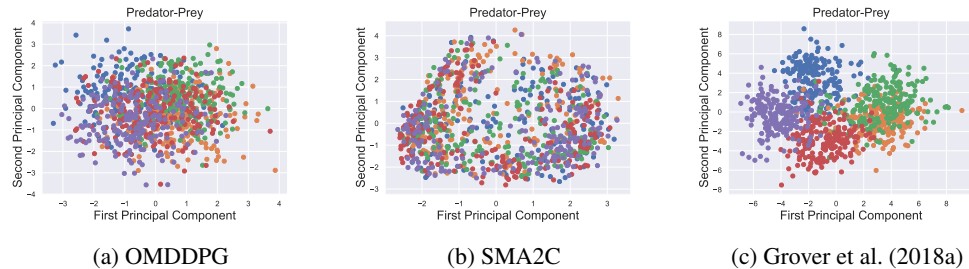

(a) OMDDPG          (b) SMA2C          (c) Grover et al. (2018a)

Figure 9: Embedding visualization from the double speaker-listener environment. The figures present 200 embeddings against each one of the five opponents and have been projected in the two first principal components from the initial ten dimensions. The embeddings presented here are from the last time step of the episode.

## D  IMPLEMENTATION DETAILS

The pseudocode for training the VAE from Section 4.2 is provided below. We consider that 1000 trajectories are provided for each one of the opponents, which are generated against trained agents. We train the VAE for 1000 epochs, using Adam (Kingma & Ba, 2014) with $10^{-3}$ learning rate.

---

**Algorithm 1** Pseudocode of the proposed VAE algorithm

---

    **for** $i = 1 : M$ **do**
        Sample an episode and compute the embedding $z \leftarrow q(\text{sample}(E_i))$
        Sample a different episode and compute the embedding $z_+ \leftarrow q(\text{sample}(E_i))$
        **for** $j = 1 : M$ **do**
            **if** $i == j$ **then**
                continue
            Sample an episode and compute the embedding $z_- \leftarrow q(\text{sample}(E_j))$
            Update VAE parameters by maximizing 8

---

We train OMDDPG for 2 million steps in all the experiments. Since the encoder of the VAE has an LSTM layer and has to be trained on sequential data, we use a modified experience replay, that enables sampling of whole episodes, which has also been used by Hausknecht & Stone (2015). DDPG algorithm requires continuous action space. However, since our experimental environments have discrete action space, we use the Gumbel-Softmax trick (Jang et al., 2016) to create differentiable samples from a discrete distribution. Additionally, we regularize the actor loss by adding the squared logits in order to prevent them from getting large values. The pseudocode of OMDDPG is presented below 2.

---

**Algorithm 2** Pseudocode of the OMDDPG algorithm

---

    **for** $e = 1 : K$ episodes **do**
        $opp \leftarrow \text{sample}(\text{opponents})$
        **while** episode is not fished **do**
            Get the observation of our agent $o$ and the observation of the opponent $o_{-1}$
            Compute the action of the opponent $a_{-1}$
            Get a sample from the encoder $z \leftarrow q(z|o_{-1}, a_{-1})$
            Compute the action $a$ of the agent using exploration
            Perform the actions in the environment and get new observations and rewards
        Store the sequences of both agents in the experience replay
        **if** $t\%\text{update\_frequency} == 0$ **then**
            Sample a batch of sequences from the experience replay
            Update the actor-critic parameters using 2 where $s \leftarrow \text{concat}(o, z)$
            Update the target networks

---

All neural networks have 2 hidden layers with ReLU (Maas et al., 2013) activation function. We use the Adam optimizer (Kingma & Ba, 2014) for all experiments. The target networks are updated with $\tau = 0.01$. The hidden dimensions for all VAE layers is 100. The parameter $\lambda$ in the VAE loss (equation 8) is always 1. We perform gradient updates every 50 time steps with a batch of 100 episodes. Table 3 summarizes the rest of the hyperparameters.

Table 3: Hyperparameters of OMDDPG, for speaker-listener (SL), double speaker-listener (DSL), predator-prey (PP) and spread (SP).

|               | SL    | DSL   | PP    | SP    |
|---------------|-------|-------|-------|-------|
| Hidden dim RL | 64    | 128   | 64    | 64    |
| Embedding dim | 2     | 10    | 10    | 8     |
| Actor lr      | 0.001 | 0.001 | 0.001 | 0.001 |
| Critic lr     | 0.001 | 0.001 | 0.01  | 0.001 |
| $\beta$       | 0.01  | 0.01  | 0.05  | 0.1   |
| Policy reg    | 0.01  | 0.001 | 0.001 | 0.001 |

We train A2C for 2 million steps in the speaker-listener environment, 5 million in the double speaker-listener, 10 million in the predator-prey and 15 million in the spread environment. A2C as an on-policy algorithm is significantly less sample-efficient compared to DDPG, and as a result, more training steps are required. The pseudocode of SMA2C is presented below 3. We subtract the policy entropy from the actor loss (Mnih et al., 2016) to ensure sufficient exploration. The loss that SMA2C minimizes can be written as:

$$
\min_{\boldsymbol{\phi}, \boldsymbol{\theta}, \boldsymbol{w}, \boldsymbol{u}} \frac{1}{2} \mathbb{E}_B [(r + \gamma V_{\boldsymbol{\phi}}(\boldsymbol{s}') - V_{\boldsymbol{\phi}}(\boldsymbol{s}))^2 - \hat{A} \log \pi_{\theta}(a|\boldsymbol{s})
$$
$$
- \delta \log p_{\boldsymbol{u}}(\boldsymbol{\tau}_{-1}|\boldsymbol{z}) + \beta D_{\mathrm{KL}}(q_{\boldsymbol{w}}(\boldsymbol{z}|\boldsymbol{\tau})\|p(\boldsymbol{z})) - b\mathcal{H}(\pi)]
$$
(13)

---

**Algorithm 3** Pseudocode of the SMA2C algorithm

Create $D$ parallel environments
$t \leftarrow 0$
**for** $e = 1 : K$ episodes **do**
    Sample $D$ opponents $opp \leftarrow \mathrm{sample}(\text{opponents})$
    **while** episode is not finished **do**
        **for** every environment in $D$ **do**
            Get the observation of our agent $o$ and the observation of the opponent $o_{-1}$
            Get a sample from the encoder $z \leftarrow q(z|o, a, r, d)$
            Compute the action $a$ of the agent using exploration
            $t \leftarrow t + 1$
            Perform the actions in the environment and get new observations, rewards and done
        **if** $t\%\text{update\_frequency} == 0$ **then**
            Gather the sequences from all environments to a single batch
            Update the actor-critic and VAE parameters using 13 where $s \leftarrow \mathrm{concat}(o, z)$

---

For the advantage computation, we use the GAE (Schulman et al., 2015b) with $\lambda_{GAE} = 0.95$. We create 10 parallel environment to break the correlation between consecutive samples. The actor and the critic share all hidden layers in all the environments except the double speaker-listener. We use the Adam optimizer (Kingma & Ba, 2014), and we clip the gradient norm to 0.5. Table 4 summarizes the rest of the hyperparameters.

Table 4: Hyperparameters of SMA2C, for speaker-listener (SL), double speaker-listener (DSL), predator-prey (PP) and spread (SP).

|  | SL | DSL | PP | SP |
|---|---|---|---|---|
| Hidden dim | 64 | 128 | 128 | 64 |
| Embedding dim | 5 | 10 | 10 | 5 |
| lr | 0.0005 | 0.0005 | 0.0005 | 0.0003 |
| $\delta$ | 0.1 | 0.5 | 1 | 0.1 |
| $\beta$ | 0.1 | 0.1 | 0.1 | 0.1 |
| $b$ | 0.1 | 0.001 | 0.01 | 0.001 |
| train frequency | 5 | 25 | 5 | 5 |

## E    AGENT GENERALIZATION GRAPHS

Figure 10 presents the agent generalization graph that was used in all experiments in this paper.

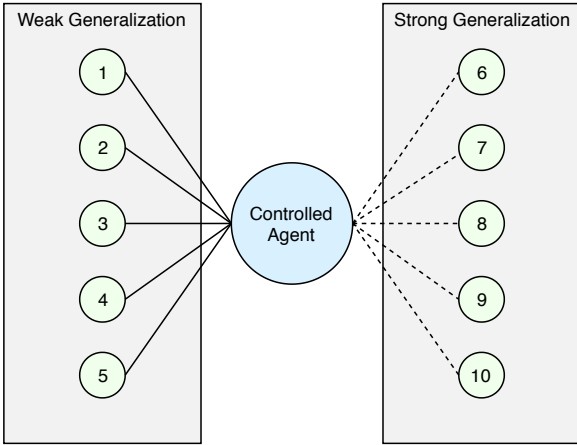

Figure 10: Agent Generalization graph. —: weak generalization, - - -: strong generalization

