# OpenReview forum: "Variational Autoencoders for Opponent Modeling in Multi-Agent Systems"
_ICLR.cc/2020/Conference — Reject_

### Official Review · AnonReviewer3 · 2019-10-23
**Official Blind Review #3**

**Rating:** 1

**Review:**

The authors propose to use VAEs to model fixed-policy opponents in a reinforcement learning setting. They use these models to augment existing RL algorithms in situations where the environment can be factorized into opponents.

I really fail to see the point of this paper. All the techniques presented in the paper are standard and the way they are put together is not particularly original. I found no specific claims about the benefits the presented approach offers over alternatives. The experiments are described from a technical perspective but I did not understand what they are actually supposed to show.

**Experience Assessment:**

I have read many papers in this area.

**Review Assessment: Checking Correctness Of Derivations And Theory:**

I assessed the sensibility of the derivations and theory.

**Review Assessment: Checking Correctness Of Experiments:**

I assessed the sensibility of the experiments.

**Review Assessment: Thoroughness In Paper Reading:**

I read the paper at least twice and used my best judgement in assessing the paper.

---

> ### Author Response · Authors · 2019-11-11
> **Reply to AnonReviewer 3.**
>
> Thank you for your review. As noted in the paper, the novelty of our work is to enable effective opponent modelling based only on local observations of our controlled agent. This is in contrast to all of the other literature on opponent modelling we are aware of, which generally assumes access to observations of opponents (e.g. their actions and/or observed states). See, for example, this recent comprehensive survey on opponent modelling methods [1]. Our work shows that VAEs can be used to achieve effective opponent modelling using only local observations, achieving comparable results to alternative methods that assume access to observations and/or actions of opponents.
>
> We emphasise that the common assumption of having access to opponent actions/observations is unrealistic for many applications, and our work is a first step in removing this assumption.
>
> [1] Albrecht SV, Stone P. Autonomous agents modelling other agents: A comprehensive survey and open problems. Artificial Intelligence. 2018

---

> ### Comment · AnonReviewer1 · 2019-11-15
> **Request for more references and grounding for your review**
>
> I agree that the techniques presented in this paper are relatively standard *but* I have yet to see a compelling study of these ideas in the context of opponent modeling in multi-agent systems and I think this paper presents some interesting experiments in this space. Do you agree that this paper has novel experiments that other techniques in the opponent modeling literature have not shown? If not, can you provide more references and details for this? Without this information, your review is not very grounded in the literature and not useful to the authors to help them contextualize their work within the community.

---

### Official Review · AnonReviewer2 · 2019-10-23
**Official Blind Review #2**

**Rating:** 3

**Review:**

The authors propose a variational autoencoding (VAE) framework for agent/opponent modeling in multi-agent games. Interestingly, as the authors show, it looks like it is possible to compute accurate embeddings of the opponent policies without having access to opponent observations and actions. The paper is well written, the methods are simple yet still interesting/informative, but there are a few questions that I find necessary to be addressed.


Methods:

1. I find the idea of learning to infer embeddings of the opponent policies from the agent's own local observations quite interesting. Intuitively, it makes sense -- since the opponent's policy effectively specifies the dynamics of the environment (from the agent's perspective), opponent's behavior must be reflected in the agent's observations. Comparing figures 1 and 2, the proposed encoder architecture also uses information about the reward (and episode termination). How critical is this information for opponent identification? Would it work without r_{t-1} and d_{t-1}?

2. Sec. 4.2: "We assume a number of K provided episode trajectories for each pretrained opponent" -- how exactly are these trajectories obtained? Similarly, how exactly are the opponents pretrained? (Self-play, hardcoded, or something else?)

3. As the authors mention, the triplet loss that discriminates between the opponents loosens the lower bound. Since the regularized objective is still a lower bound, I wonder if the triplet loss can be re-interpreted/expresses as a specific prior on the opponent model?


Experiments:

1. Sec. 5.1: to understand the effect of opponent modeling, it would be nice to see how baselines perform in this setup against a randomly picked opponent (otherwise, the curve in Fig. 3-c is not informative). I suggest the following baselines: tit-for-tat (hardcoded), a couple of classical learning algorithms for iterated games (e.g., policy hill-climbing, WoLF), an agent that learns using policy gradients but without opponent embeddings. Without any baselines, Sec. 5.1 seems like a sanity check which just shows that the implementation works unless I am missing something.

2. Sec. 5.3: (1) Why is mutual information between the approximate posterior q and the prior p makes sense as the policy embedding quality metric here? (2) Could you intuitively (or formally) justify the fact that the triplet loss degrades MI metric? Right now, this is stated as a fact but not justified. (3) It looks like Grover et al. (2018a) used deterministic trajectory encoders; how exactly is MI measured in that case?

3. If I understand correctly from Fig. 4, SMA2C (which uses local information) underperforms as compared to the methods that use opponent trajectories in 6/8 cases. To me, this somewhat confirms the point opposite to what the authors claim -- local observations, while containing some information about the opponent, are still inferior. Also, having baselines that do not use opponent embeddings on the charts of Fig.4 would help understand the contribution of opponent modeling.

----

I acknowledge reading the author's response, which addressed some of my questions/concerns to some extent. However, I believe that while estimating accurate embeddings of the opponent behavior from the agent's observations only is interesting, the approach has limitations, and I feel those are not studied in-depth enough (e.g., as a reader, I would like to understand if and when I should use the proposed approach and expect it to work). My assessment of the paper stays the same.

**Experience Assessment:**

I have published one or two papers in this area.

**Review Assessment: Checking Correctness Of Derivations And Theory:**

I assessed the sensibility of the derivations and theory.

**Review Assessment: Checking Correctness Of Experiments:**

I carefully checked the experiments.

**Review Assessment: Thoroughness In Paper Reading:**

I read the paper thoroughly.

---

> ### Author Response · Authors · 2019-11-11
> **Reply to AnonReviewer 2.**
>
> We would like to thank you for your review and your comments. We are delighted that you found the method interesting. Below, we try to address your comments about the methods and the experiments.
>
> Methods:
> 1) Note that we provided an ablation study in Appendix B of the paper. We have moved the ablation study from the Appendix to the main text and added plots for more environments.
> In most environments, the model performs similar without using r_{t-1} and d_{t-1} for weak generalization. For strong generalization, the performance in speaker-listener and double speaker-listener decreases.
>
>
> 2) The trajectories of the opponents are gathered against agents which were trained using MADDPG. Details can be found in Appendix D.
>
> In the speaker-listener environment, we control the listener, and we create ten different speakers using different communication messages for different colors (hardcoded). In the double speaker-listener, which consists of two agents that have to be both listener and speaker simultaneously, we control the first agent. We create a diverse set of opponents that have different communication messages similar to speaker-listener (hardcoded), while they learn to navigate using the MADDPG algorithm (training), with different initial random seeds. In the predator-prey environment, we control the prey and pretrain the three other agents in the environment using MADDPG with different initial parameters (training). Similarly, in spread, we control one of the agents, while the opponents are pretrained using MADDPG (training). Note that these details are included in Section 5.2 of the paper.
>
>
> 3) While we do not yet have a mathematically sound argument to support this, we tend to agree with your opinion. Intuitively, we believe that the triplet loss could probably be interpreted as a discrete prior, such as a Bernoulli prior on the latent variables. This could be an interesting investigation in subsequent work.
>
> Experiments:
> 1) Our intention, in this case, was indeed to show that our method can achieve effective opponent modelling with only local observations. In this case, we assume that there are two opponents with fixed policies. Following the reviewer's suggestion, we have added one of the requested baselines (A2C, a policy gradient method) in the new Figure 3-c. The other training-based baselines have exactly the same behavior as A2C. Note that in this experiment, we assume that the training agents have access only to local observations and not the actions of the opponents.
>
> 2) (1) We use mutual information as a means to evaluate the learned representation. We practically compute the MI between the embedding distribution q(z|x) and the prior on the labels which is uniform, not the prior on the latent variables which an isotropic Gaussian. This is a common way to evaluate representations in the literature of representation learning. Practically, high MI could be interpreted as a high statistical dependence between the embeddings and the opponents' identities. For example, the example of Figure 3-b has high MI which is close to the upper bound. By counting the MI, we want to measure whether the encoder learns any relationship between the agent labels and the embeddings. One of the most powerful solutions that we could provide to the problem described in section 4.1, which however uses strong assumptions, is to concatenate the observation of our agent along with a one-hot vector that represents the identity of each opponent.
> (2) We discovered a small mistake in our experiments. The value of MI in table 1 was incorrect. We have updated the paper accordingly. The triplet loss forces the linear separability between the embeddings of different opponents, which practically increases the MI. We thank the reviewer for pointing us to this error.
> (3) Note that we estimate the MI from samples and not analytically from the distributions. Even though the representations, that Grover et al use, are deterministic there is some underlying distribution and the embeddings could be considered as samples from this distribution.
>
> 3) We emphasise that our goal is not to outperform methods that use opponent information; instead, our goal is to enable opponent modelling when only local information is available. It should be expected that one can perform better if opponent information is used.

---

### Official Review · AnonReviewer1 · 2019-10-24
**Official Blind Review #1**

**Rating:** 6

**Review:**

This paper proposes a reasonable and natural way of modeling
opponents in multi-agent systems by learning a latent space
with a VAE. This latent space will ideally learn the strategy
that the opponent is playing to inform the agent's policy.
With fixed opponents, the results across many tasks are convincing.

My one concern with this modeling approach is that it will start
breaking down if the opponents are *not* fixed as this
potentially makes the agent more exploitable.
The opponents could learn to send adversarial sequences to
the opponent model that make it appear like they are playing
one strategy but then they could change strategies at
a critical point where it is too late for the agent to recover
or perform optimally.
This type of exploitability has been explored in the game
theory community in [1,2] and the references therein.

[1] Ganzfried, S., & Sandholm, T. Game theory-based opponent modeling in large imperfect-information games. AAMAS 2011.
[2] Ganzfried, S., & Sandholm, T. Safe opponent exploitation. TEAC 2015.

**Experience Assessment:**

I have read many papers in this area.

**Review Assessment: Checking Correctness Of Derivations And Theory:**

I did not assess the derivations or theory.

**Review Assessment: Checking Correctness Of Experiments:**

I assessed the sensibility of the experiments.

**Review Assessment: Thoroughness In Paper Reading:**

I read the paper at least twice and used my best judgement in assessing the paper.

---

> ### Author Response · Authors · 2019-11-11
> **Reply to AnonReviewer 1.**
>
> We would like to thank you for your review and your comments. Our work is indeed focused on modelling opponents with fixed policies.
>
> Our current paper is a first study on using opponent modelling with VAEs when only local observations are available, and for this first step, we decided to assume opponents with fixed policies. Dealing with opponents whose policies change over time is indeed an important open problem, and this direction is part of our future work.
>
> We have added citations to the references that you provided. Thank you for bringing them to our attention.

---

> > ### Comment · AnonReviewer1 · 2019-11-15
> > **Reviewer Response**
> >
> > Thanks for the response and clarifications here. I've read through the other reviews and maintain my original score of a weak accept as I think there are some interesting ideas in here, even if the resulting agents are still exploitable from a game-theoretical perspective and we don't yet understand how to use the approach with changing opponents

---

### Author Response · Authors · 2019-11-11
**Note for all AnonReviewers**

We would like to thank all the reviewers for the feedback that they provided. We uploaded a newer version of our work where we address some of the issues raised by the reviewers as well as fix some typos.

---

### Decision · Program_Chairs · 2019-12-19

**Decision:**

Reject

**Comment:**

The present work addresses the problem of opponent modeling in multi-agent learning settings, and propose an approach based on variational auto-encoders (VAEs). Reviewers consider the approach natural and novel empirical results area presented to show that the proposed approach can accurately model opponents in partially observable settings. Several concerns were addressed by the authors during the rebuttal phased. A key remaining concern is the size of the contribution. Reviewers suggest that a deeper conceptual development, e.g., based on empirical insights, is required.